# Ribosomal 18S rRNA base pairs with mRNA during eukaryotic translation initiation

Franck Martin[1,*], Jean-François Ménétret[2,3,4,5,*], Angelita Simonetti[1,*], Alexander G. Myasnikov[2,3,4,5], Quentin Vicens[1], Lydia Prongidi-Fix[1], S. Kundhavai Natchiar[2,3,4,5], Bruno P. Klaholz[2,3,4,5] & Gilbert Eriani[1]

Eukaryotic mRNAs often contain a Kozak sequence that helps tether the ribosome to the AUG start codon. The mRNA of histone H4 (*h4*) does not undergo classical ribosome scanning but has evolved a specific tethering mechanism. The cryo-EM structure of the rabbit ribosome complex with mouse *h4* shows that the mRNA forms a folded, repressive structure at the mRNA entry site on the 40S subunit next to the tip of helix 16 of 18S ribosomal RNA (rRNA). Toe-printing and mutational assays reveal that an interaction exists between a purine-rich sequence in *h4* mRNA and a complementary UUUC sequence of helix h16. Together the present data establish that the *h4* mRNA harbours a sequence complementary to an 18S rRNA sequence which tethers the mRNA to the ribosome to promote proper start codon positioning, complementing the interactions of the 40S subunit with the Kozak sequence that flanks the AUG start codon.

[1] Architecture et Réactivité de l'ARN, Centre National de la Recherche Scientifique (CNRS) UPR9002, Institute of Molecular and Cellular Biology (IBMC), Université de Strasbourg, 15 rue René Descartes, 67084 Strasbourg, France. [2] Department of Integrated Structural Biology, Centre for Integrative Biology (CBI), IGBMC (Institute of Genetics and of Molecular and Cellular Biology), 1 rue Laurent Fries, 67404 Illkirch, France. [3] CNRS UMR 7104, 67404 Illkirch, France. [4] Institut National de la Santé et de la Recherche Médicale (INSERM) U964, 67404 Illkirch, France. [5] Université de Strasbourg, 67081 Strasbourg, France. * These authors contributed equally to this work. Correspondence and requests for materials should be addressed to B.P.K. (email: klaholz@igbmc.fr) or to G.E. (email: g.eriani@ibmc-cnrs.unistra.fr).

In eukaryotes, the start codon is identified through base-triplet scanning by the initiator-tRNA bound 40S ribosomal subunit (43S complex), starting from the usually m$^7$G-capped 5′ end until the correct AUG start codon is found and the 48S initiation complex is formed. At least 13 initiation factors are involved in translation initiation which results in the formation of the 80S initiation complex on joining of the 60S ribosomal subunit[1–5]. To ensure the fidelity of translation initiation, the start codon is usually located in the context of a Kozak sequence (A/G)CCAUGG (ref. 6) and contains a purine in position −3 and a G in position +4. Variations of the Kozak sequence can lead to initiation at downstream AUG triplets by leaky scanning[7]. However, deviations from this classical model exist, for example, viral mRNAs that contain 5′ untranslated region (UTR) internal ribosomal entry sites (IRES) often require only a subset of the initiation factors to hijack the ribosome, as visualized by several cryo-EM structures[8–13]. Histone H4 mRNA (*h4*) combines canonical features (cap-dependent translation) with viral strategy (lack of scanning). It contains a three-way junction (TWJ) with the unusual property of stalling engaged 80S ribosomes when cap-dependent translation is repressed[14]. The TWJ is located 19 nucleotides downstream from the AUG codon, and is flanked by a weak Kozak sequence (with a U in position +4) and a double stem-loop structure called eIF4E-sensitive element (4E-SE) (Supplementary Fig. 1)[14]. These specific RNA structures tether the translation machinery directly on the first AUG initiation codon of *h4* mRNA, regardless of the presence of a second in-frame initiation codon. The lack of scanning appears to favour high expression levels of histone H4 protein during S-phase of the cell cycle, which is relevant for chromatin organization, but the regulatory mechanism is unknown. Here we localize the folded *h4* mRNA TWJ domain on the rabbit ribosome using cryo-EM and show by toe-printing and mutational analysis that *h4* mRNA exhibits shortly after the start codon a sequence complementary to the 18S rRNA sequence that helps mRNA binding and proper AUG positioning.

## Results

### Structure of the 80S ribosome assembled on histone *h4* mRNA.

Mouse *h4* mRNA/rabbit 80S complexes were assembled in rabbit reticulocyte lysate and stalled in the initiation state by cycloheximide and hygromycin B that prevent the elongation at translocation steps. Complexes were pulled from the extracts by affinity purification[15] and analysed by cryo-EM. Rabbit reticulocyte lysates mimic the full complexity of the *in vivo* environment, and provide all required tRNAs besides translation factors for efficient assembly. However, the process also limited to some extend the resolution of the structure due to stronger sample heterogeneity, which could only in part be addressed by particle sorting. The cryo-EM structure of the predominant subpopulation nevertheless reached ∼10 Å resolution, which allowed localizing the *h4* TWJ on the 80S ribosome. Further high-resolution refinement provided better features on the ribosome but not on the *h4* region probably due to multiple conformations (see Methods). It shows that *h4* forms a folded, repressive structure bound to the 40S subunit at the mRNA entry site (Fig. 1a). The cryo-EM map was interpreted by fitting the atomic model of the human ribosome derived from high-resolution cryo-EM[16]. The structure contains an initiator tRNA accommodated in the peptidyl (P) site and a ternary complex of eEF1A-tRNA localized in the factor-binding site (Fig. 1a) reminiscent of a late 80S translation initiation complex in which codon recognition has occurred. A separate sub-class also shows the post-translocation complex with eEF2 (see Methods and Supplementary Fig. 2). The 5′ extremity of *h4* is positioned close

to ribosomal proteins eS26 and eS28 as confirmed by chemical crosslinking experiments performed with *h4* harbouring a periodate-oxidized cap (Supplementary Fig. 3). The role of these proteins is supported by a recent study that showed how the IRES of hepatitis C virus (HCV) mimics a bacterial Shine–Dalgarno (SD)–anti-SD structure and interacts with eS26 and eS28 to facilitate mRNA loading and tRNA binding into the P-site[17]. The mRNA extends towards the mRNA entry site at position 26. A large additional density (by comparison with the empty ribosome, Supplementary Fig. 4) is located in this region, reminiscent of the folded TWJ RNA element. It is embedded between helix h16 (18S rRNA) and ribosomal proteins uS2, uS3 and eS10 of the 40S beak (Fig. 1b). The structure of the 80S ribosome complex with a deletion mutant of *h4* comprising nucleotides 1–142 (*h4*$_{1–142}$, rather than 377 nt; Supplementary Fig. 4) confirms that the density corresponds to the 5′ region of *h4*. Its size accounts for the ribosome-interacting TWJ part while the 3′ region of the mRNA is disordered.

### Interaction between ribosomal helix 16 and *h4* mRNA.

The binding of the *h4* mRNA at the tip of helix h16 (18S rRNA) suggests that a direct interaction between *h4* mRNA and the rRNA exists. This 18S rRNA region comprises an apical ($_{540}$UUUC$_{543}$) tetraloop in which the four nucleotides are often found to be flipped out in various ribosomal structures. To identify the possible nucleotides interacting with the 18S rRNA we probed the mRNA structure by nucleotide substitution and monitored binding to the ribosome by reverse transcriptase assays ('toe-printing'). A toe-print was detected at position +17 (numbering starting on the A of the AUG codon, or *h4* nt 27), 3 nt upstream of the TWJ domain (Fig. 2a). Interestingly, nts 26 to 30 ($_{26}$AAGGG$_{30}$) could base pair with nts of h16 tetraloop ($_{540}$UUUC$_{543}$) to form a putative interaction site at the entrance of the mRNA channel. Such interaction would be consistent with the distance between the mRNA on the AUG in the P site and the TWJ on helix h16 as shown by mRNA modelling (Fig. 1c). This prompted us to mutate these nts of *h4* and check whether ribosome positioning was modified. Single mutants of nts 26 to 30 were constructed and tested. They all exhibited a toe-print at position +17, but in addition also one at position +26 with an intensity inversely proportional to the one at position +17 suggesting that these nucleotides critically influence mRNA positioning and ribosome assembly (Supplementary Fig. 5). In fact, toe-prints at position +26 indicate slippage of the ribosomes towards an out-of-frame AUG-like codon (G$_{21}$U$_{22}$G$_{23}$). We further combined mutations in double mutants and confirm ribosome slippage on the AUG codon, especially with mutants (28–29) and (29–30) (Supplementary Fig. 5). A triple mutant (27–28–29) exhibited a more drastic effect with toe-prints being spread over positions +17, +18 and +19 (Fig. 2a). These new shifts indicate that the mutated mRNA strand is less constrained in the mRNA channel and up to 2 extra nts can enter the mRNA cleft to give rise to the +18 and +19 stops (Fig. 2a,b). These results show that interactions between the initiator region of the mRNA and the 18S rRNA are required to avoid ribosome slippage over the AUG start codon. Along the same lines, nucleotide deletion downstream of the AUG induces a shift of the toe-print to position +16 (Supplementary Fig. 5). This shows that the interaction with h16 is strong enough to stretch the mRNA by one nt in the mRNA channel. However, when 2 nts are deleted, the toe-print moves back to +17 suggesting loss of the h16–*h4* interaction. Consistently, 1 nt deletion in the triple mutant (27–28–29) did not induce any shift of the toe-print position that would indicate the mRNA stretching. This further confirms that the interaction of h16 with residues (27–28–29) is

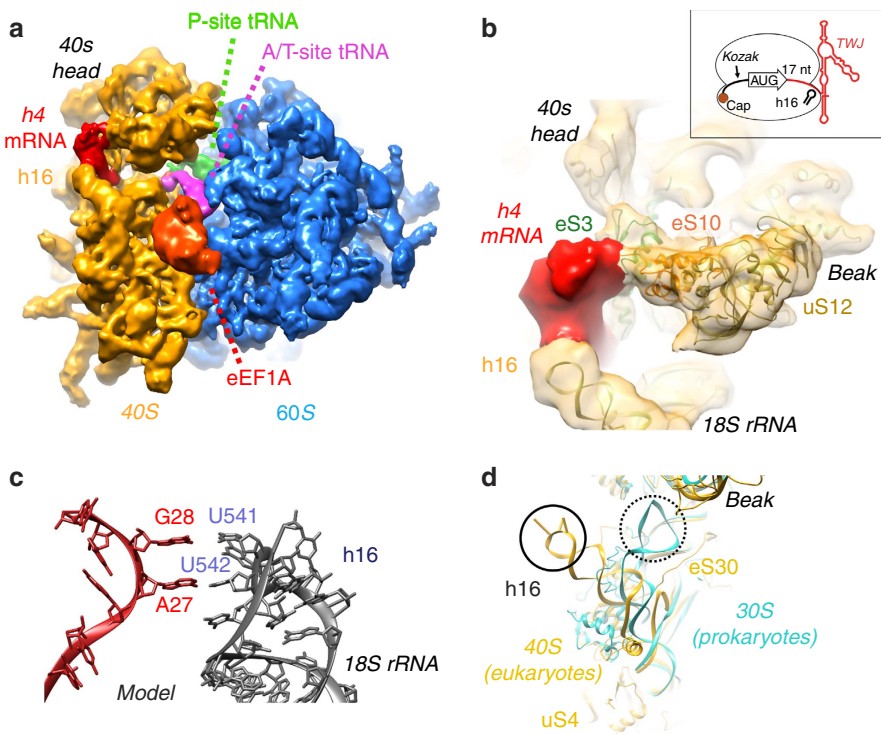

**Figure 1 | Localization of the histone *h4* mRNA on the 80S ribosome.** (**a**) Overview of the *h4*/80S complex stalled in the pre-translocation state with *h4* (red), eEF1A (red), A/T-site tRNA (magenta), P-site tRNA (green), 60S ribosomal subunit (blue) and 40S ribosomal subunit (orange). (**b**) *h4* mRNA is inserted between the tip of ribosomal helix h16 (18S rRNA) and proteins uS3 and eS10. (**c**) Model of the *h4* mRNA interactions with the apical loop of 18S rRNA helix h16. (**d**) Superimposition of eukaryotic and prokaryotic ribosomes highlighting the structural difference at the level of helix h16, creating a site in eukaryotes for mRNA binding (solid circle); eukaryote-specific protein eS30 in part takes the place of h16 in prokaryotes.

contributing to mRNA binding (Supplementary Fig. 5). Together, these experiments establish that interactions between h16 and *h4* mRNA exist and are critical for optimally positioning the mRNA on the ribosome (Fig. 1d). To evaluate the significance of this interaction on poly-ribosome formation, polysome profiles of translation extracts programmed with wild-type *h4* and the triple mutant (27–28–29) were examined. Compared with the triple mutant, the wild-type *h4* was more efficient in ribosome assembly and translation. Indeed, 42.5% of the mRNA of the triple mutant were not assembled with ribosomes versus 31.5% for the wild-type *h4*. In addition, wild-type *h4* exhibited more polysomes ( + 11%), suggesting that it is more efficiently translated (Fig. 2c).

**Binding of yeast 40S by a compensatory mutant of *h4*.** To address the role of the h16 tetra-loop residues, we performed additional binding assays of *h4* mRNA with 40S ribosomes. To demonstrate the base pairing between the mRNA and rRNA, we set out to identify compensatory mutations that restore the binding of *h4* mRNA to a mutated h16 tetra-loop. As the production of mutated rabbit ribosomes is very challenging, we focused the experiment on purified yeast 40S subunits, which naturally exhibit a variation of the tetraloop of h16 and do not bind *h4* mRNA (Fig. 3). We then set about finding new *h4* mRNA mutants that would generate yeast 40S binding. Significant binding of the 40S particles was obtained with ($G_{29}U$) mutant, which allows formation of an additional $U_{29}$:$A_{540}$ pair instead of the $G_{29}$:$A_{540}$ pair (Fig. 3). To check whether the newly formed $U_{29}$:$A_{540}$ pair is the essential element of the ribosome:mRNA interaction, we tested the binding of two additional mRNA mutants. A first one was a triple mutant ($U_{27}U_{28}U_{29}$) that exhibited the restoring $U_{29}$:$A_{540}$ pair. A second one was a triple mutant ($U_{26}U_{27}U_{28}$) displaced by one nt that kept the

non-functional $G_{29}$:$A_{540}$ pair (Fig. 3). Both mutants did not bind yeast ribosomes. This result shows that the $U_{29}$:$A_{540}$ pair cannot lead to ribosome binding in the absence of the pairings on the 5′ side. We cannot exclude the possibility that the conformation of the yeast tetraloop is quite different than the rabbit tetraloop. Indeed, tetraloops starting with A are typically not well structured, in contrast to those starting with U. This is the case of the (AUUC) tetraloop of yeast h16 (ref. 18), in contrast to the (CUUU) tetraloop of rabbit h16 (ref. 19). Therefore, formation of an extra $U_{29}$:A pair could rearrange the yeast tetraloop structure and favour binding of 40S subunits. Altogether, these results validate the importance of the h16 interaction site also for the yeast ribosome, and suggest that this binding mode may be widely used in the eukaryotic kingdom.

**Discussion**

Taken together, the present data uncover the concept of base pairing between 18S rRNA sequence and eukaryotic mRNAs to facilitate ribosome positioning on the start codon, complementing the stabilizing role of the Kozak consensus sequence that flanks the AUG start codon. Structural and functional data reveal that the key regulatory site for this is the tip of eukaryotic helix h16 which can base pair with the *h4* sequence preceding the TWJ. This additional interaction may compensate for weak or deficient Kozak consensus sequences at + 4. This tethering mechanism provides specificity for the formation of translation initiation complexes on the first start codon of *h4* and explains why slippage on a second start codon does not occur. By directly forming base-pair interactions with the tip of the ribosomal h16, it increases the general affinity for the small subunit and correctly localizes the ribosome on the *h4* start codon thus preventing scanning. According to the wobble rules, residue U has the ability

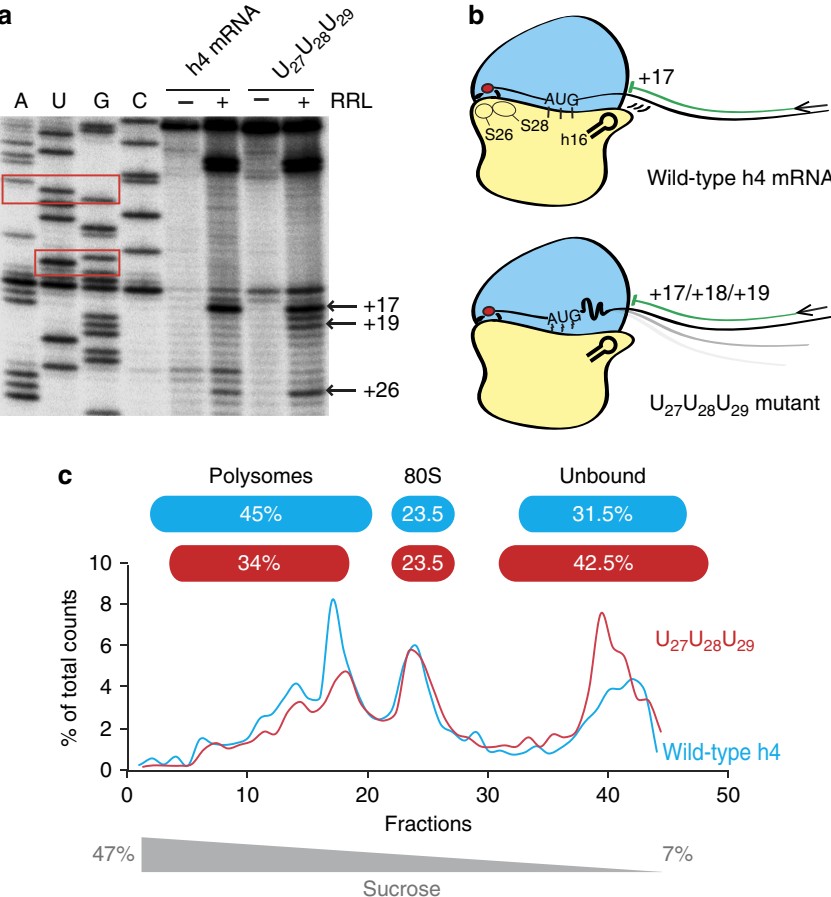

**Figure 2 | Ribosome toe-prints and polysome fractionation with *h4* mRNA and mutant.** (**a**) Initiation complexes were assembled in RRL extracts in the presence of cycloheximide and hygromycin B to stall the initiation complexes on the AUG codon. Reaction samples were separated on 8% denaturing PAGE together with the appropriate sequencing ladder (shown on the left). AUG initiation codon and GUG codon are boxed. Toe-print positions were numbered starting on the A of the AUG codon (+17 position corresponds to *h4* nt 27). (**b**) Model of *h4* interaction within the 80S ribosomal particle. Accurate positioning of *h4* mRNA results from interactions with helix h16 from 18S rRNA. Mutation of nts 27–29 induces toe-print shifts to +18 and +19 indicating that the mRNA is not accurately maintained into the mRNA channel. (**c**) Polysome fractionation of translation extracts programmed with wild-type *h4* mRNA and derived triple mutant. Ribosome assembly and translation was studied in RRL programmed with 5′-end radiolabelled m7G-capped *h4* mRNA. Unblocked translation extracts were separated on 7–47% sucrose gradients and radiolabelled mRNAs were detected by Cerenkov counting. The graph represents the radioactivity in the different fractions expressed as a percentage of the total radioactive counts. The positions of polysomes, 80S and free mRNA are indicated. The sums of counts measured in polysomes, 80S particles and not assembled (unbound) are indicated in the blue and red bars for wild-type and triple mutant, respectively.

to base pair with A and G residues. Therefore, the complexity of base pairing with the UUUC sequence is increased, suggesting that many other mRNAs may be assisted by the interaction with h16 in a similar way. In addition, the presence of the TWJ-folded domain locks the ribosome in a pre-translocation conformation to stabilize the base pairing interactions. In that position, the TWJ of *h4* also competes with DHX29 (ref. 20) a critical helicase for the scanning mechanism[21]. This observation is consistent with the absence of scanning of the short *h4* 5′UTR (ref. 14). The N-terminal domain of Hbs1 protein (part of the no-go decay complex[22]) also binds at this particular place[23]. The discovery of mRNA interactions with specific bases of the 18S rRNA appears to be a mechanism reminiscent of that observed in bacteria at the level of the SD interactions at the 3′ end of the 16S rRNA that help recruiting mRNAs to the 30S initiation complex. However, the interaction site observed in the eukaryotic complex is completely different because it corresponds to a eukaryote-specific sequence insertion in the 18S rRNA (tip of h16), which is oriented differently and extends by ∼50 Å as compared with bacterial ribosomes (Fig. 1d) to create a landing platform for

pre-binding the mRNA at the entry site of the mRNA channel (Fig. 1b). This allows formation of stabilizing interactions of the mRNA with the ribosome that promote the formation of the 48S initiation complex, illustrating how temporarily repressive folded elements of cellular mRNAs can guide the ribosome to favour their own translation. The study thus brings in a new concept regarding the mode of interaction of mRNAs with specific structural elements of a eukaryote-specific site on the 40S subunit, the general significance being comparable to that of Kozak and Shine–Dalgarno sequences. An interesting question to address in future studies is whether this specific 18S rRNA interaction exists with other eukaryotic mRNAs. mRNA:rRNA interactions are more documented in viruses. For instance, sequences in the adenovirus mRNA complementary to 18S rRNA facilitate shunting by base pairing to 40S ribosomal subunit[24]. A base pairing between hepatitis C virus and 18S rRNA is also required for IRES-dependent translation initiation[25]. Several studies reported similar interactions with cellular mRNAs. These include reports of mRNA interactions between a plant ribosomal protein mRNA (RPS18) and the 18S rRNA[26],

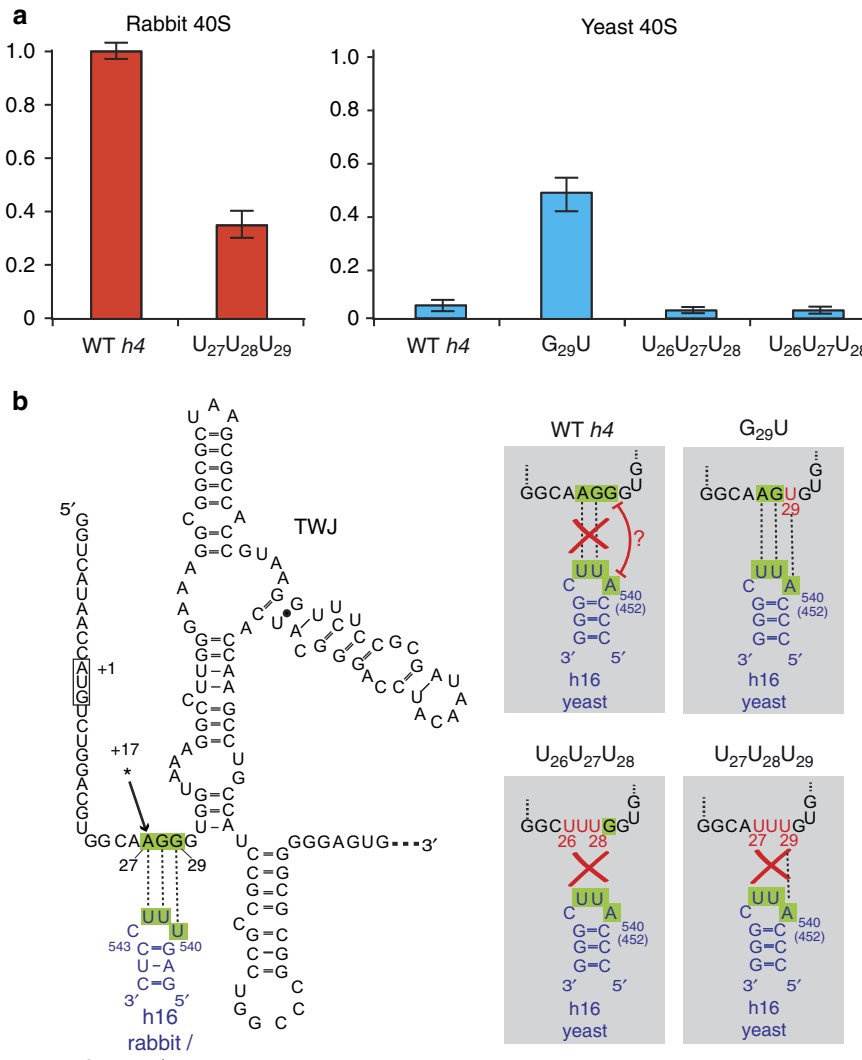

**Figure 3 | Ribosome binding on *h4* mRNAs.** (**a**) Histogram showing 40S subunit binding on *h4* mRNAs. Binding was studied on sucrose gradient with radiolabelled m7G-capped *h4* mRNA. Samples were separated on 7–47% sucrose gradients, and complexes with 40S particles were counted in Cerenkov mode. Binding values were normalized to wild-type *h4* binding with rabbit 40S particles. Values represent the average of three technical replicates. Errors bars representing the variability of data are shown. (**b**) Secondary structure of the 142 first nucleotides of murine histone *h4* mRNA. The structure contains three helices connected by a TWJ followed by a stem-loop structure. The initiation codon is boxed. The black star indicates the location of the +17 ribosome toe-print. Partial helix 16 (h16) from yeast and mammalian (rabbit, human and mouse) are drawn in blue; nts numbering corresponds to rabbit sequence (rabbit 540 = yeast 452). Mutated *h4* mRNAs tested with 40S subunits from yeast are shown in the grey insets.

a ribosome shunt involving the 5′ leader of the *Gtx* mRNA and 18S rRNA interaction[27], mRNA:rRNA base pairing in translation of the *Gtx* and FGF2 (fibroblast growth factor 2) mRNAs[28,29]. Altogether, data obtained by all these model systems suggest that base pairing with 18S rRNA could be also relevant to develop improved eukaryotic protein expression systems that bypass scanning and would imitate highly translated mRNA.

## Methods
**Sucrose gradient analysis.** The protocol used to prepare the complexes with 5′-labelled capped mRNA (250,000 c.p.m. per tube) is basically the same as described hereafter (section toe-prints analysis). The complexes were separated on 7–47% linear sucrose gradients in buffer (25 mM Tris-HCl (pH 7.4), 75 mM KCl, 0.5 mM MgCl$_2$, 1 mM DTT, 1 mM cycloheximide). The reactions were loaded on the gradients and spun (37,000 r.p.m. for 2.5 h at 4 °C) in a SW41Ti rotor. Gradients were fractionated and analysed by Cerenkov counting.

**Chemical crosslinking of the 5′ cap and immunoprecipitation.** *h4* mRNA, radiolabelled at the level of the G of the cap, was periodate-oxidized before assembling initiation complexes in reticulocytes extracts in the presence of various inhibitors or cycloheximide. Five micrograms of purified and radiolabelled (250,000 c.p.m.) capped *h4* mRNA were incubated for 2–3 h at 0 °C in 250 µl of 100 mM sodium acetate (pH 5.3), 10 mM EDTA, 0.2 mM sodium periodate. Then, glycerol was added to 2% final concentration. After 10 min incubation at room temperature, the mixture was phenol extracted twice and ethanol precipitated. The RNA pellet was dissolved in 10 µl of water. One microlitre of oxidized RNA was incubated with 4 µl of rabbit reticulocyte lysates (RRL), in 10 mM HEPES-KOH (pH 7.6), 1 mM ATP, 75 mM KCl, 1 mM DTT, 1 mM Mg(Ac)$_2$ and 1 mg ml$^{-1}$ cycloheximide, or 2 mM AMP-PCP or 2 mM GMP-PNP or 1 mM m7GDP in a final volume of 10 µl. After 10 min incubation at 30 °C, 1 µl of 0.2 M NaBH4 was added and incubation was extended for 2–3 h at 0 °C. Then, RNA was digested by 1 µl of RNase A (Roche) for 30 min at 37 °C and samples were fractionated on SDS–polyacrylamide gel. Ribosomal protein eS28 was further immunoprecipitated with specific antibodies coupled to MagnaBind Protein G beads according to the manufacturer's instructions (Thermo Scientific).

**Ribosome toe-printing.** Prior to the formation of rabbit 80S/mouse *h4* initiation complexes, untreated RRL (Green Hectares, USA) were incubated for 5 min at 30 °C and 20 min in ice in a buffer containing 1 U ml$^{-1}$ RNaseOUT Recombinant Ribonuclease Inhibitor (Invitrogen), 75 mM KCl and 0.5 mM MgCl$_2$. Then, RRL were incubated in the presence of 1.3 mM puromycin at 30 °C during 5 min. To lock the ribosome at translation initiation, RRL were then incubated for 3 min at

30 °C in the presence of a mix of 1 mg ml$^{-1}$ cycloheximide and 0.5 mg ml$^{-1}$ hygromycin B blocking the translocation of the peptidyl-tRNA from the A to the P site of the ribosome[30]. Finally, formation of initiation complexes was obtained by adding histone h4 mRNA at a final concentration of 500 nM and incubating for 5 min at 30 °C. Then, ribosome complexes (15 µl) were mixed with an equal volume of ice-cold buffer A containing 20 mM Tris-HCl (pH 7.5), 100 mM KAc, 2.5 mM Mg[Ac]$_2$, 2 mM DTT, 1 mM ATP and 0.25 mM spermidine. Toe-print experiments were adapted from refs 14,31. An ultracentrifugation of the reaction mixture step was performed at 337,000g in a S100AT3 rotor (Sorvall-Hitachi) at 4 °C for 1 h to separate ribosomal complexes from the non-ribosomal fraction. Then, ribosomal pellets were dissolved in 30 µl buffer A complemented with the same translation inhibitor and analysed by primer extension using AMV reverse transcriptase and a primer complementary to nts 91–110 of h4 (ref. 14).

**Sample preparation for cryo-EM.** 80S/h4 and 80S/h4$_{1-142}$ ribosome complexes were prepared as described previously[15]. Briefly, mouse h4 mRNA was ligated to a biotinylated DNA oligonucleotide and bound to streptavidin-coated beads. Then, rabbit 80S ribosomes were assembled on the beads coated with the bait, stalled at the post-initiation step, washed, and released from the beads by enzymatic DNase I cleavage of the DNA moiety[15]. First, the chimeric mRNA–DNA bait harbouring a biotin molecule at its 3′ end was constructed in one step ligation catalysed by T4 DNA ligase[15]. Then, the chimeric molecule (50 µg in 50 µl) was incubated with 150 µl of pre-washed streptavidin-coated beads (MagSI-STA 600—MagnaMedics) in binding buffer (100 mM potassium phosphate, pH 7.2, 150 mM NaCl) for 30 min at room temperature, and washed with water. In parallel, nuclease-untreated RRL (100 µl, Green Hectares) was added of 50 mM KAc, 0.4 U µl$^{-1}$ of RNasin (Promega) and 100 µM of each of the 20 amino acids in a total volume of 200 µl. The mix was incubated at 30 °C for 5 min and then chilled 20 min on ice. The immobilized hybrid h4 mRNA was then incubated 5 min at 30 °C with the RRL translation mixture in presence of 1 mg ml$^{-1}$ cycloheximide and 0.5 mg ml$^{-1}$ hygromycin B. After an additional 3 min incubation on ice in the presence of 8 mM Mg(Ac)$_2$ and 5 min incubation on ice in 200 µl of buffer (2 mM DTT, 100 mM KAc, 20 mM HEPES-KOH (pH 6.5), 2.5 mM Mg(Ac)$_2$, 1 mM ATP, 0.1 mM GMP-PNP and 0.25 mM spermidine), the complexes were sequentially washed with ice-cold buffers containing 250 mM KAc (twice), then 500 mM KAc (once) and 50 mM KAc (three times)[15]. At the end, the stalled h4/80S complexes were eluted in 100 µl of elution buffer (50 mM KAc, 20 mM HEPES-KOH (pH 6.5), 1 mM DTT, 10 mM Mg(Ac)$_2$, 1 mM CaCl$_2$) by 10 U of RQ1 RNase-free DNase (Promega), during 30 min at 37 °C. The eluted complexes were collected by centrifugation for 1 h at 108,000 r.p.m. ( = 680,000g) in a S140AT rotor (Sorvall-Hitachi) at 4 °C. Ribosomal pellets were resuspended in 20 mM HEPES-KOH (pH 6.5), 0.2 mM EDTA, 50 mM KAc, 1 mM Mg(Ac)$_2$, 1 mM DTT) to a concentration of 10 A$_{260}$ U ml$^{-1}$.

Empty 80S ribosomes were purified from nuclease-untreated RRL by centrifugation at 37,000 r.p.m. in a SW41Ti rotor for 2.5 h at 4 °C through 7–47% linear sucrose gradient in buffer containing 25 mM Tris-HCl (pH 7.5), 50 mM KCl, 5 mM MgCl$_2$ and 1 mM DTT. After gradient fractionation, fractions containing 80S ribosomes were centrifuged at 108,000 r.p.m. (S140AT Sorvall-Hitachi rotor) for 1 h at 4 °C, then the ribosomal pellet was dissolved in 80S/h4 complex resuspension buffer (20 mM HEPES-KOH (pH 7.6), 0.2 mM EDTA, 10 mM KCl, 1 mM MgCl$_2$, 1 mM DTT).

**Data collection.** A volume of 2.5 µl of freshly prepared 80S ribosome complexes, at 0.2–0.5 mg ml$^{-1}$, were applied to 300 mesh holey carbon Quantifoil 2/2 grids (Quantifoil Micro Tools, Jena, Germany), blotted with filter paper from both sides for half a second in the temperature- and humidity-controlled Vitrobot apparatus (FEI, Eindhoven, Netherlands, $T = 10$ °C, humidity 95%, blot force 8, blot time 0.5 s) and vitrified in liquid ethane pre-cooled by liquid nitrogen. Data were collected on the in-house spherical aberration (Cs) corrected Titan Krios S-FEG instrument (FEI, Eindhoven, Netherlands) operating at 300 kV acceleration voltage and at a nominal underfocus of $\Delta z = -0.6$ to $-4.5$ µm using a second-generation back-thinned direct electron detector CMOS (Falcon II) 4,096 × 4,096 camera and automated data collection with EPU software (FEI, Eindhoven, Netherlands). The camera was set up to collect seven frames, plus one total exposure image; total exposure time was 1 s with a dose of 60 ē Å$^{-2}$ (3.5 ē Å$^{-2}$ per frame) using a nominal magnification of × 59,000 resulting in 1.1 Å pixel size at the specimen level (images were coarsened by 2 for further processing). Data for the empty 80S, 80S/h4$_{1-142}$ and preliminary 80S/h4 ribosome complexes were collected on the in-house Polara Tecnai F30 electron microscope using a first-generation direct electron detector CMOS (Falcon I) 4,096 × 4,096 camera using a magnification of × 59,000 with a pixel size of 1.36 Å.

**Image processing.** Stack alignment of the Titan Krios data was performed before particle picking, which included seven frames and a total exposure image (total eight images in the stack), using the whole image motion correction method[32]. Thereafter, an average image of the whole stack was used to pick 146,821 particles semi-automatically using EMAN2 Boxer[33] and RELION[34], and the contrast transfer function of every image was determined using CTFFIND3 (ref. 35) in the RELION workflow. Particle sorting was done by two-dimensional classification

resulting in 48,952 particles. Further three-dimensional classification resulted in five classes with 2,822, 7,893, 6,786, 3,412 and 5,363 particles (total 26,276 particles). Classes 1, 3, 4 and 5 looked similar with h4 present in the folded state, and A- and P-site tRNAs and eEF1A; these classes were merged for structure refinement (18,383 particles). Class 2 contained elongation factor eEF2, P/E-site tRNA and no density for h4, which corresponds to the elongated complex in which the h4 mRNA is unfolded and tRNA is already translocated (Supplementary Fig. 2). This complex is typical of cycloheximide inhibition that happens after a first translocation step by blocking tRNA$^{Met}$ into the E-site[36]. Hygromycin B that typically prevents the translocation induced by eEF2 (refs 37–39) was probably not bound in this complex. Sorting was also applied to the 80S/h4$_{1-142}$ data, revealing the same mass of density for the 5′ core domain of h4, but with the 40S in different conformations (Supplementary Fig. 6). The resolution was estimated in Relion at 0.143 FSC[34], indicating an average resolution of 10.2 Å (Supplementary Fig. 7). Map interpretation was done using Chimera[40] and Coot[41] starting from our human ribosome atomic model[16] which was fitted by rigid body and real-space refinement using Phenix[42]. Figures were prepared using the software Chimera[40] and Pymol (The PyMOL Molecular Graphics System, Version 1.5.0.4 Schrödinger, LLC.; DeLano, 2006).

**h4 RNA modelling.** Helix h16 from the 3.65 Å cryo-EM structure of O. cuniculus (PDB ID 3JAG) was superimposed onto h16 in the ribosome structures from O. cuniculus (PDB ID 4UJE) and H. sapiens (PDB ID 4UG0) used for map fitting. The mRNA from a partial 48S preinitiation complex in S. cerevisiae (PDB ID 3J81) was edited according to sequence differences and fitted in density using Coot and the UCSF Chimera package. The structure data file (.sdf) for 7-methyl-guanosine-5′-triphosphate was retrieved from PDB entry 3AM7 (ref. 43). The .pdb file generated from the .sdf file by eLBOW[44] in Phenix[42] was fitted in density using Chimera. Based on a comparative analysis of various tetraloops, we selected the tetraloop from h16 in a S. cerevisiae translation initiation complex (PDB ID 3JAM) to model base pairs involving U541–G28 and U542–A27. Geometry of the mRNA and h16 were regularized in Coot.

**Data availability.** The experimental map is available from the Electron Microscopy Data Bank (EMDB) under accession code EMD-4049. All other relevant data are available from the authors upon request.

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

## Acknowledgements

We thank L. Schaeffer for technical support, as well as P. Auffinger and L. d'Ascenzo for helpful discussions on tetraloops. This work was supported by ANR-2011-svse8-02501 (MITIC project), the European Research Council (ERC Starting Grant N_243296 TRANSLATIONMACHINERY) and the Centre National pour la Recherche Scientifique (CNRS). The electron microscope facility was supported by the Alsace Region, the FRM, INSERM, CNRS and the Association pour la Recherche sur le Cancer (ARC) and by the French Infrastructure for Integrated Structural Biology (FRISBI) ANR-10-INSB-05-01, and Instruct as part of the European Strategy Forum on Research Infrastructures (ESFRI).

## Author contributions

F.M. conceived and performed the mutation analysis, crosslinks, 40S binding assays and polysome analysis. A.S. performed toe-print experiment and conducted samples preparation and optimization for cryo-EM study and interpretation of the cryo-EM map. J.F.M. performed cryo-EM data acquisition and image processing. J.F.M., A.M. and S.K.N. performed EM structure refinement and model building. Q.V. modelled *h4* mRNA and human 18S ribosomal helix h16. L.P.F. developed initial protocols for the assembly and purification of complexes. B.P.K. and G.E. supervised the study. All authors analysed the data. B.P.K. and G.E. wrote the manuscript with inputs from A.S. and F.M.

## Additional information

**Competing financial interests:** The authors declare no competing financial interests.

