## [Peer Review File · Nature Communications]

Reviewers' Comments:

Reviewer #1 (Remarks to the Author)

We are all taught (or teach) that one of the major differences between translation initiation in prokaryotes and eukaryotes is that in the former, mRNA/rRNA base pairing interactions are central to the former, but that this does not apply in the latter case. Indeed, it is argued that this is why eukaryotic initiation requires so many trans-acting factors. However, histone mRNAs have always been considered to be strange exceptions to the rules, particularly because they lack the polyA tails and have extremely short 5' UTRs. Here, using a combination of cryo-EM and molecular biology approaches, Martin and co-workers provide very strong evidence supporting the hypothesis that base pairing interactions between the human h4 mRNA and helix 16 of the small subunit rRNA function to position the correct AUG codon in the decoding site. This is a quite remarkable discovery because it upends our conceptual understanding of the fundamental differences between eukaryotes and prokaryotes.

Cryo-EM reconstructions of rabbit 80S ribosomes complexed with the human h4 mRNA identified the possible interaction between 26AAGGG30 of the h4 mRNA and the 540UUUC534 tetraloop of h16. Some very nice experiments, including ribosome toeprinting (showing that the both sequence and spacing are required to properly position the ribosome at the correct AUG), and a polysome analysis (showing that the AAGGG sequence is required for optimal translation of the h4 mRNA in vitro) are performed. One of the critical experiments involves showing that mutating nucleotides 27 - 29 to UUU strongly interferes with the ability of rabbit ribosomes to bind the h4 mRNA. In contrast, since this change renders the sequence complementary to the h16 tetraloop in yeast ribosomes, the mutant mRNA is now able to much more strongly interact with yeast ribosomes. This raises the question: does the yeast mRNA harbor sequence complementary to its cognate h16 sequence? The authors should look. Additionally, with this in mind, I would like to suggest a genetics experiment in which an mRNA harboring the 5' sequence of the rabbit h4 mRNA (coupled to a reporter of choice) is expressed yeast cells. One would predict that UUU mutant would be much more highly expressed in yeast as compared to the wild-type sequence. Ideally, one would also like to mutate the rRNA side of the interaction. While yeast based systems do exist, the considerations that the experiments would 1) very high risk (there is a strong probability that the mutant ribosomes would be quickly degraded by the non-functional ribosome decay apparatus) and 2) so far beyond the scope of this lab, lead this reviewer to not-recommend that this line of research be required.

Lastly, while novel, the work could benefit from some additional referencing. mRNA:rRNA interactions in eukaryotes are pretty well documented in viruses. There are also a couple of reports of such interactions with cellular mRNAs. These include reports of mRNA interactions between a plant ribosomal protein mRNA and the 18S rRNA (Vanderhaeghen et al 2006 FEBS Lett 580: 2630), a ribosome shunt involving an mRNA: rRNA interaction (Chappell et al, 2006 PNAS 103: 9488), mRNA:rRNA base pairing in translation of the Gtx and FGF2 mRNAs (Panopolos and Mauro JBC 2008 283: 33087, and Dresios et al NSMB 2006 13: 30).

Minor issues (language)

1. At the end of the abstract the authors write that "...an 18S rRNA sequence exists to tether the mRNA to the ribosome through complementary base pairing and promote start codon positioning...". The way that this is written, it implies that the 18S rRNA evolved in response to the need for the h4 mRNA's AUG codon to be properly positioned in the ribosomal decoding center. This is logically backwards because the ribosome existed well before histones (and their mRNAs) came into existence. The correct way to write this would be to switch the order of the nouns, e.g. "...the h4 5' leader harbors a sequence complementary to an 18S rRNA sequence which tethers the mRNA to the ribosome to promote proper start codon positioning...". This linguistic problem is also

found at the end of the first paragraph of the introduction.

2. In the abstract, line 11. 'a purine', not 'an purine'.

3. P. 3, line 1. Substitute 'identified' for 'found'. Also in this paragraph (and in general throughout) the authors like to use very long sentences containing multiple ideas. This makes the work hard to read. Suggestion: consider using a period instead of a comma.

4. P. 3 line 8. Substitute 'and' for 'which'.

5. P. 3, line 11. Substitute 'often' for 'which'

6. p 4 line 4. 'to obtain' is very awkward. Suggest re-wording. Same line, substitute 'extent' for 'extend'.

7. P 6, paragraph 2, end of line 7, substitute 'for' for 'into'.

8. P 6, paragraph 2, line 9. 'Explains' instead of 'explain'.

9. P. 7 line 19. "formation of" instead of 'forming'

Reviewer #2 (Remarks to the Author)

The structures of histone mRNAs are atypical in several ways, and there is good evidence that translation initiation of these mRNAs is also atypical. Some of the authors of the present manuscript presented data in 2011 (Martin et al, Mol Cell 41, 197) that histone H4 mRNA contains several interesting structural elements important for noncanonical initiation, most notably (1) a three-way junction that binds the cap; and (2) a double stem-loop structure that binds factor 4E. They suggested that these presumably histone-specific elements recruit 4E in a novel way, and also tether histone mRNAs to the small subunit, resulting in efficient translation during S phase. Here the authors have performed structural and biochemical experiments to characterize more fully the mechanism of noncanonical initiation on histone H4 mRNA. Relatively low-resolution (~10 Å) cryoEM data show a large mass of density on the solvent side of the beak, in a location consistent with the expected location of the three-way junction. This mass of density is observed to interact with a eukaryotic-specific portion of helix 16. This observation suggested the possibility that a single-stranded portion of H4 mRNA may base-pair with single-stranded nucleotides in the hairpin loop of helix 16. To test this hypothesis, the authors performed a variety of biochemical experiments on wild-type and mutant forms of H4 mRNA. In particular, toeprinting and binding affinity experiments were performed to test whether the helix 16 apical loop does in fact base-pair with a portion of H4 mRNA.

I am not convinced that the authors have proven that actual base-pairing occurs between helix 16 and H4 mRNA.

While the cryoEM data certainly show a large mass of density for the three-way junction in the expected location, the resolution is far too low to prove the existence of base-pairing between helix 16 and histone H4 mRNA.

The toeprinting of wild-type and mutant forms of H4 mRNA also cannot establish base-pairing, and does not add much to the paper.

The most convincing way to establish a base-pairing interaction would be to disrupt the interaction by mutagenesis, and then show rescue by a compensatory mutation. The authors did make an attempt similar to this strategy. It is not a simple matter to mutate the ribosomal RNA of higher eukaryotes, so the authors used yeast 40S subunits, which differ somewhat (1 nt) in the apical loop of helix 16. This sequence difference is predicted, according to the authors' model, to change a G-U pair to a G-A pair. Binding of H4 mRNA to these yeast 40S is indeed greatly reduced, though a reduction in binding could arise from many different mechanisms. However, mutation of the H4 mRNA so as to change the predicted G-A pair to a U-A pair does result in a significant rescue of binding affinity. (I should add that a G-A pair may not always be less stable than a G-U pair, or a U-A pair!)

While certainly consistent with their helix 16 - H4 mRNA base-pairing model, this essentially single

result is not in my opinion sufficient to prove the base-pairing interaction. I would encourage the authors to pursue additional experiments to prove their hypothesis. Either a higher-resolution structure or a larger body of biochemical data could prove (or disprove) their model. In the absence of proof of the interaction, the paper is significantly less interesting. Nevertheless I would encourage the authors to persevere, as such proof could establish an entirely new flavor of noncanonical translation initiation.

Point-by-point response to the referees

Response to Referee 1:

Referee 1 also suggested an elegant complementation assay in yeast with a mammalian H4 mRNA fused with a reporter. In fact, we already tried to fuse H4 with reporter genes in the past. However, we failed in obtaining an active translation. Now we know that this inactivation of H4 translation is the consequence of the tethering mechanism that we previously described (Martin et al, Mol Cell 41, 197) which involves structural elements well ordered in the coding region of the mRNA. We believe that the fusion damages the tethering mechanism by some structural or steric effect that prevents translation of the fusion. We understand that having a translation system with an active H4-reporter fusion would have allowed performing the genetic experiment suggested by reviewer 1 (complementing assay with yeast ribosome). Unfortunately, we couldn't establish an active fusion between H4 protein and a reporter. This would have validated in vivo the binding result that we got in vitro with purified yeast 40S subunit (Figure 3).

Concerning the question: does the yeast mRNA harbor sequence complementary to its cognate h16 sequence? The answer is no. In yeast, there are two histone H4 mRNAs. Both exhibit the AGG sequence found in mammalian H4 mRNA. The AGG sequence is overlapping two codons (**AAG** GGU, bold underlined) coding for two nearly invariant amino acid residues (Lys6 and Gly7) in the histone H4 family. In yeast, it has been shown that Lys6 is a reversible acetylation site necessary for promoter activation. It was shown that replacement of Lys6 by another amino acid causes activation to be repressed. Therefore, these observations about the functional importance and conservation of these amino acids may explain the nucleotide conservation of the AGG triplet in yeast. Theoretically, the AAG codon encoding Lys6 could have evolved towards the second Lys codon (AAA); but this would not favor the interaction with helix 16. Optimizing the interaction with yeast helix 16 would require a U at the first position of the codon for Gly7, inducing a Gly->Cys substitution (probably deleterious). Together, these observations suggest that in yeast, the conservation of the AGG sequence might result from a pressure onto the peptide sequence of the N-terminus of histone H4 rather than a selection pressure for keeping a translation mechanism.

Moreover, in this regard, we would like to add that yeast histone mRNAs are quite different than those found in mammals. They are polyadenylated and exhibit long 5' and 3'UTRs. Unpublished experiments performed in our lab showed that translation initiation on yeast histone H4 mRNA occurs according to the classical scanning mechanism involving both 5' UTR and 3' polyA tail. This might be correlated with the highly degenerated sequence of yeast H4 mRNAs when compared with highly conserved metazoan H4 sequences. This divergence suggests that the key RNA structural elements that drive tethering of the translation machinery in mouse H4 mRNA have been lost as well as the resulting tethering mechanism.

Referee 1 suggested that our work could benefit from some additional referencing. Therefore, we added the following sentences and references:

"mRNA:rRNA interactions are more documented in viruses. For instance, sequences in the adenovirus mRNA complementary to 18S rRNA facilitate shunting by base pairing to 40S ribosomal subunit²⁴. A base pairing between hepatitis C virus and 18S rRNA is also required for IRES-dependent translation initiation²⁵. Several studies reported similar interactions with cellular mRNAs. These include reports of mRNA interactions between a plant ribosomal protein mRNA (RPS18) and the 18S rRNA²⁶, a ribosome shunt involving the 5' leader of the *Gtx* mRNA and 18S rRNA interaction²⁷, mRNA:rRNA base pairing in translation of the *Gtx* and FGF2 (fibroblast growth factor 2) mRNAs^{28,29}."

24. Ryabova, L. A., Pooggin, M. M. & Hohn, T. Viral strategies of translation initiation: ribosomal shunt and reinitiation. *Prog. Nucleic Acid Res. Mol. Biol.* **72**, 1–39 (2002).

25. Matsuda, D. & Mauro, V. P. Base pairing between hepatitis C virus RNA and 18S rRNA is required for IRES-dependent translation initiation in vivo. *Proc. Natl. Acad. Sci. U. S. A.* **111**, 15385–9 (2014).
26. Vanderhaeghen, R. *et al.* Leader sequence of a plant ribosomal protein gene with complementarity to the 18S rRNA triggers in vitro cap-independent translation. *FEBS Lett.* **580**, 2630–6 (2006).
27. Chappell, S. A., Dresios, J., Edelman, G. M. & Mauro, V. P. Ribosomal shunting mediated by a translational enhancer element that base pairs to 18S rRNA. *Proc. Natl. Acad. Sci. U. S. A.* **103**, 9488–93 (2006).
28. Panopoulos, P. & Mauro, V. P. Antisense masking reveals contributions of mRNA-rRNA base pairing to translation of Gtx and FGF2 mRNAs. *J. Biol. Chem.* **283**, 33087–93 (2008).
29. Dresios, J., Chappell, S. A., Zhou, W. & Mauro, V. P. An mRNA-rRNA base-pairing mechanism for translation initiation in eukaryotes. *Nat. Struct. Mol. Biol.* **13**, 30–4 (2006).

We also corrected the minor issues.

1. At the end of the abstract the authors write that "...an 18S rRNA sequence exists to tether the mRNA to the ribosome through complementary base pairing and promote start codon positioning...". The way that this is written, it implies that the 18S rRNA evolved in response to the need for the h4 mRNA's AUG codon to be properly positioned in the ribosomal decoding center. This is logically backwards because the ribosome existed well before histones (and their mRNAs) came into existence. The correct way to write this would be to switch the order of the nouns, e.g. "...the h4 5' leader harbors a sequence complementary to an 18S rRNA sequence which tethers the mRNA to the ribosome to promote proper start codon positioning...". This linguistic problem is also found at the end of the first paragraph of the introduction.

This has been corrected in both abstract and introduction

2. In the abstract, line 11. 'a purine', not 'an purine'. done
3. P. 3, line 1. Substitute 'identified' for 'found'. Also in this paragraph (and in general throughout) the authors like to use very long sentences containing multiple ideas. This makes the work hard to read. Suggestion: consider using a period instead of a comma. done
4. P. 3 line 8. Substitute 'and' for 'which'. done
5. P. 3, line 11. Substitute 'often' for 'which' done
6. p 4 line 4. 'to obtain' is very awkward. Suggest re-wording. Same line, substitute 'extent' for 'extend'. The whole paragraph has been rewritten.
7. P 6, paragraph 2, end of line 7, substitute 'for' for 'into'. done
8. P 6, paragraph 2, line 9. 'Explains' instead of 'explain'. done
9. P. 7 line 19. "formation of" instead of 'forming' done

Response to Referee 2:

The main issue raised by referee 2 concerned the demonstration of base pairing between the mRNA and the ribosomal RNA. We agree that clear demonstration of base pairing would require high cryo-EM resolution structure, in order to “see” the base interactions. However, despite our efforts, the complexes assembled in translation extracts were too heterogeneous to reach high resolution. Particles sorting could partially solve the heterogeneity problem but the resulting final map at 10 angstroms resolution could not show formally the base pairing in greater details. However, we collected a body of evidences that strongly suggest that interactions exist between the mRNA and 18S RNA. First, we performed mutational analysis followed by toeprint analysis. The results showed that changing nts 27, 28, 29, individually or in combinations of double and triple mutants induced changes of the ribosome toe-prints. Generally, the expected +17 toe-print is decreased whereas a new one resulting from a ribosome slippage appeared at +26 (corresponding to ribosome assembly on a GUG codon). The triple mutant 27-28-29 exhibited toe-prints spread over +17, +18 and +19

which can only be explained by the loss of an interaction with the ribosome and by the entry of the mRNA into the ribosome channel of two extra-nucleotides (the toe-print is considered to map the mRNA entry channel on the ribosome). To emphasize this result, we modified figure 2. We added a cartoon that illustrates the mRNA shift in the ribosome mRNA channel. We hope that the figure will strengthen this important result of the manuscript and facilitate understanding.

Second, we performed complementation assays with yeast ribosomes. Using yeast ribosomes unable to bind H4 mRNA, we isolated a mutant of H4 mRNA that became able to bind the yeast ribosome. The mutant mapped nt 29 of H4 mRNA as a critical element for the interaction. Changing the G29:A pairing (inactive) to a Watson Crick U29:A pair led to mRNA binding to the ribosome. Remarkably, this single nt substitution restored the possibility to form 3 pairs between the mRNA and helix16. However, as reviewer 2 points out, a G:A pair is not always less stable than a U:A pair. Consequently, the binding of U29:A mutant pair should not necessarily be improved compared to the G29:A pair.

In fact, stability of G:A pairs is very variable. At least nineteen different non Watson Crick G:A pairings have been described in 3D structures (Leontis, Stombaugh & Westhof NAR 2002, 30, 3497-531). In many examples, the distance between the riboses of interacting nts is larger for G:A pairing than conventional pairing (Leontis et al., 2002). This suggests that the geometry of the G:A pair (WT H4 with yeast h16: no binding) could be significantly different than the G:U (WT H4 with human h16: efficient binding) or the U:A pair (restoring mutant G29U with yeast h16). A simple assumption could be that G29:A pair clashes with adjacent nts and acts as a negative element precluding productive interaction of nt 27 and 28 with helix 16. This could explain the absence of binding of the mRNA. With this in mind, we performed and added to the manuscript new binding data of two triple U mutants: U(26-28) and U(27-29) (fig 3). One mutant theoretically allows formation of a G29:A interaction and the other one a U29:A interaction. Both mutants did not bind yeast ribosomes. This shows that nt 29 alone cannot drive ribosome binding. It needs the pairing with the adjacent 5' nts as seen in G29U mutant and further confirms the specific interaction with these nts.

These data have been added to the new figure 3. We also modified the drawing of the putative pairings in the new figure 3. We added a repressive pictogram for the G29:A interaction to suggest the repressive effect of this interaction on the binding of yeast 40S.

Another more likely assumption to explain the binding of yeast 40S would be that the tetraloop of yeast h16 exhibits a quite different conformation. Commonly, tetraloops starting with A are typically not structured, in contrast to those starting with U. This is the case of the tetraloop of yeast h16 (Pdb 4V88 and 4U3M, both at 3 angstrom resolution), in contrast to the CUUU tetraloop of rabbit h16 (Pdb 3JAG, 3.65 angstrom resolution). Therefore, formation of an extra U29:A pair could stabilize yeast h16 loop structure by engaging 3 nts in pairings with h4 mRNA and this could explain the binding of 40S subunits that we observed.

We have also revised the end of the manuscript results accordingly to these new ideas and assumptions.

Reviewers' Comments:

Reviewer #1 (Remarks to the Author)

The authors have effectively addressed the issues raised in my initial review. This is a very interesting contribution to the field.

Reviewer #2 (Remarks to the Author)

In my review of the first version of this manuscript I expressed reservations about how well the authors were able to provide evidence for base pairing between the H4 mRNA and helix 16 of the 40S subunit.

As the cryoEM data were of insufficient resolution to address this issue, the authors examined the interaction of wild-type and a mutant mRNA with yeast 40S, whose helix 16 differs in sequence from mammalian. In my first review, I was concerned that the evidence for actual basepairing seemed to rely on data from just the wild-type and a single mRNA mutant.

The authors have added data from two additional H4 mRNA mutants, and I agree that these data are consistent with the H4 mRNA-helix 16 basepairing model.

I still would not say that the authors have rigorously proven the existence of this basepairing, but the additional data, together with the biochemical and low-resolution structural data presented here, make their model convincing.

I would like to congratulate the authors on a very interesting and important new model for noncanonical translation initiation.